# Political participation and voluntary associations : A hypergraph case study

**Amina Azaiez** [1] *, **Robin Salot**[2,3]

1 Sorbonne Economics Center, Paris 1 Panthéon Sorbonne University, Paris, France, 2 Laboratoire d'Anthropologie Sociale, Ecole des Hautes Etudes en Sciences Sociales, Paris, France, 3 Centre d'Etude des Mouvements Sociaux, Ecole des Hautes Etudes en Science Sociales, Paris, France

* amina.azaiez@univ-paris1.fr

**Data Availability Statement:** All relevant data are held in a public repository at https://github.com/aminaazaiez/Politics-and-associations/tree/main/data.

## Abstract

Civic organizations, ranging from interest groups to voluntary associations, significantly influence policy formation in representative democracies. This work presents a local case study that examines the relationship between voluntary associations and local political institutions in a village with nearly two thousand residents. Traditionally, sociologists' approaches have focused on individual characteristics such as age, gender, or socio-professional status. In this study, we analyze social interactions between members of organizations modeled through a hypergraph. Specifically, we model interactions as hyperedges that correspond to activities proposed by organizations and involve the individuals who participate in those activities. Our analysis reveals a community-based structure, where members of similar types of organizations tend to interact more frequently. To quantify 'political participation,' we introduce an interaction-based measure that extends degree centrality. We also introduce the 'diversity coefficient' as an extension of degree centrality to capture an individual's ability to participate in activities composed of members from different communities. Among other centrality measures, we find that the diversity coefficient is the most significant factor in explaining political participation among members of associations.

## Introduction

The interdependence between social participation and political involvement has long been studied in social and political sciences. One of the earliest and most well-known works on the subject is Tocqueville's *Democracy in America*, which emphasizes the importance of social participation and intermediary associations in democracy. Some organizations, such as 'intermediary organizations' or 'interest groups' (e.g., trade unions or business organizations), are explicitly linked to politics because they take part in institutionalized decision-making procedures, and their function is to influence those decisions in favor of the group they represent [1–3]. On the other hand, other kinds of voluntary associations, such as educational associations or art and sports clubs, do not have an explicit political function. Nonetheless, it has been argued that even these 'apolitical' associations spur political participation [4–6].

Historically, social scientists have sought to explain the political commitment of citizens by appealing to personal characteristics such as age, gender, social status and membership in

**Funding:** The author(s) received no specific funding for this work;.

**Competing interests:** The authors have declared that no competing interests exist.

associations [4, 5]. Others, drawing on theories of social-psychology and communication, argue that the social environment and micro-interpersonal interactions might be key factors for understanding political involvement and political orientations [7–10]. These two schools of thought—one focusing on individual characteristics and the other on micro-interactions—offer distinct but complementary insights into the understanding of political participation. Our work aligns more closely with the second methodological approach, relying on network theory. However, the originality of our work lies in considering the whole as a socio-political ecosystem. Thus, we model both social and political interactions through a single network.

Indeed, this literature usually limits the study of "political participation" to voting patterns and participation rates in national elections. However, social participation and political participation can be seen as a two-way flow. The concept of "politics" can be extended to all activities that serve to organize social groups, without being limited to the act of voting. In this sense, the boundaries of politics can be expanded by taking it out of its purely professional field [11–14]. From this perspective, it is not necessary to have an established position in political spaces to effectively mobilize political techniques. For example, techniques such as the ability to speak in public, to distribute individual tasks with authority or horizontality for the organization of collective activities, to establish strategies for seizing power or contesting the authorities instituted by democratic tools, etc., can be found in both the associative and political spheres. Beyond these technical similarities, these associations can also influence politicians at the city level through their frequent contact with city council members, for example, when organizing social events or setting up social services such as childcare or help for the elderly. Additionally, politicians may see these associations as potential channels for contacting and convincing their members, thereby becoming vehicles for electoral legitimization [11]. Without going so far as to homogenize the two spheres [15], we define political participation as access to local politics. More formally, we measure the frequency with which people participate in activities involving local elected officials.

In this paper, we study social interactions in a village of approximately two thousand residents to reveal the interdependency between social and political participation. We focus our study on members of voluntary non-profitable associations and elected officials through the interactions occurring within and between these organizations. We model these face-to-face interactions through a network, more precisely, a hypergraph where the hyperedges correspond to activities proposed by the organizations and involve the individuals who participate in those activities. At the macroscale, these data can be seen as a snapshot of the underlying social fabric of a village. At the micro-scale, these interactions can serve to understand complex social behavior that permits access to local politics.

Through our study, we find that the network is organized into distinct communities and that the structure of these communities overlies the structure of organizations within the village. To better understand the interplay between social and political participation, we introduce two extensions of the degree centrality that take into account the composition of the hyperedges on top of their cardinality: one measures the "political participation" of agents, the other, called "diversity coefficient", captures the ability to participate in heterogeneous activities gathering individuals from different communities. We compare the correlation between the political participation and various centralities: the diversity coefficient, and others from the literature. Our findings suggest that, for members of associations, engaging in heterogeneous activities is the most effective way to gain access to politics.

## Related work and hypotheses

Most previous studies investigating the link between social networks and political behavior have focused on egocentric networks of survey respondents, examining how social interactions

with "alters" may influence the political behavior of the "ego". Various influential processes have been explored in this research. These include the frequency of political discussions, the level of political knowledge possessed by alters, the homogeneity of political preferences among alters [16, 17], and the social diversity of the network [6, 18]. These surveys had the advantage of being addressed to many respondents with diverse social capital. As a result, findings tend to be more representative and generalizable at the scale of the concerned country.

However, this method has many theoretical drawbacks. First, the ego-networks typically consist of only a few friends or acquaintances, usually ranging from three to six individuals. Knoke [16] acknowledges that a thorough investigation of social processes requires comprehensive network data over time and direct measures from all participants. However, due to time constraints, surveys often only collect information on a few close contacts of respondents. Connections between these contacts and second-degree connections (friends of friends) are usually neglected. This results in a star-shaped network where the ego is at the center and the alters are connected to the ego, but not to one another. This creates an implicit assumption of a one-directional causal relationship, where alters influence ego but not vice versa.

A second important limitation of this method is the restriction of ego-networks to core ties when asking respondents with whom they talk about 'important matters'. Scholars have argued for the importance of weak ties, [19–21] especially for the diffusion of information and access to new knowledge. Granovetter [19] theorized that communities are social groups with strong ties and high internal density. He argued that connecting these communities requires weak ties, as it is unlikely to have a close friend outside one's core community. Therefore, only weak ties can serve as bridging ties, connecting disjoint communities. The importance of these bridging weak ties for dynamic diffusion processes within social groups is clear.

Applying this theoretical concept to explain the level of political participation, bridging ties should enhance political knowledge and understanding, thereby increasing participation in political activities [22–24]. The results of Crenson's 1978 [22] study in Baltimore seem to support this theory. He compared community associations in six neighborhoods to reveal the factors behind the associations' successful operation. He found that residents with loose-knit neighborhood friendship ties were more likely to be informed about community associations and that the latter were more responsive to their interests. Another comparative study by Ohlemacher [23] reached similar results. His analysis of the organization membership affiliation networks of two protest groups in West Germany shows that successful mobilization correlates with the presence of structural bridging links.

Moreover, Ohlemacher insists on the need for these links, as their absence in the weak mobilization group leads to network fragmentation. This result is also highlighted by Eveland and Kleinman [24] in their study of 25 voluntary groups in a university. Comparing their general and political discussion networks, they found that political discussion networks are more likely to be broken down into subcomponents than general discussion networks. These empirical studies suggest that in networks with 'structural holes' [20], individuals who bridge distinct social groups play a crucial role, provided the network maintains a community structure. Siegel [25] emphasizes this by proposing a model for political participation viewed as a dynamical process on complex networks. He showed that in village networks composed of cliques connected by weak ties, increasing the number of weak ties significantly boosted political participation. Conversely, in other types of network structure, adding weak ties could reduce political participation, depending on the distribution of internal motivations.

Beyond the structural importance of bridging ties, Ohlemacher [23] argued that they are vectors for social heterogeneity, linking a variety of people with different values, experiences and socio-structural background. Even though some studies claim that network heterogeneity decreases individual political participation due to "cross-pressures" [26], more recent studies

[6, 18] showed that political and social diversity foster political participation. Diverse networks and exposure to disagreement encourage individuals to reevaluate their knowledge and learn about alternative perspectives.

This research aims to investigate the relationship between the level of political participation of individuals and their position in the network. As explained in the Context section, the studied rural municipality is more accurately described as a village rather than an urban city. Consequently, we expect the network to be dense, reflecting a social environment where 'everybody knows everybody else.' However, given that our data concerns specific organizations and that individuals likely interact more frequently with members of their own organization, we expect a community structure within the hypergraph. Based on the previous conceptual discussion, we propose the following hypotheses:

H1: The hypergraph has a community structure based on the structure of the organizations.

H2: People located at the interface of these communities have more access to local politics.

Now that we have set the groundwork for the sociological conceptual framework of our study, let us emphasize the choice of using hypergraphs for modeling social interactions. In our case, face-to-face interactions correspond to individuals participating in the same activity. To model these interactions, a natural representation is a hypergraph rather than a graph, since the activities under consideration possibly involve more than two individuals. Unlike traditional graphs, hypergraphs offer the possibility to represent hyperedges linking more than two nodes. Projecting this network into its clique expansion by artificially creating links between each pair of nodes present in a given hyperedge results in an intentional loss of information.

This choice is not merely for convenience; recent studies reveal the importance of preserving higher-order interactions in various structural analyses such as centrality measurement or community detection algorithms [27, 28]. The interest in using this type of modeling is particularly significant when studying diffusion in social systems, such as the adoption of innovations or norms, the spread of rumors, and opinion dynamics. Empirical studies [29, 30] demonstrate that social contagion phenomena rely on higher-order interactions involving non-linear dynamics that cannot be reduced to the sum of dyadic interactions. Models for opinion dynamics also provide theoretical insights in this direction [31, 32].

This paper aims to contribute modestly to this literature in two ways. First, we propose measures that extend the degree centrality of nodes by taking into account the composition of hyperedges. Second, in the S1 File, we perform a comparative analysis of the results obtained using a community detection algorithm for the hypergraph and its clique expansion.

## Materials and methods

### Context

Since our study was conducted in a particular social and geographical context, it is important to specify this context so that other studies with similar or contradictory results could draw conclusions about the generalizability of these results or the sources of inconsistencies.

The case study village is located in Seine et Marne, France. Its center, built at the gateway to a valley, contrasts with the scattering of its hamlets established on the agricultural plateaus. The population of this small village doubled between 1968 and 2022, growing from 1000 to 2000 inhabitants. After the retirees, who represent 25.4% of the population, service sector employees (20.2%) and workers (15.4%) are the most represented groups. Caught up in the daily migration typical of rural areas, these employees largely abandon associative places. As is

often the case in rural areas, these associations, like the municipality, are mostly run by retired people.

There is a notable structural similarity between these two types of organization: each association is made up of a general assembly of its members, which elect a board of directors, which in turn elects a bureau consisting of a president, a treasurer and a secretary. In the same way, in each municipality, a municipal council is elected by direct universal suffrage by the inhabitants, which in turn elects a mayor and his deputies. In addition to this organizational resemblance, associations and the town hall are frequently in contact: the town hall solicits associations during national events (national day, heritage festival, etc.), associations request funding from the town hall, elected officials are invited to the general meetings of associations, and elected officials participate in associative activities.

The village under consideration is part of a community of municipalities. A community of municipalities is an EPCI (Public Establishment for Intermunicipal Cooperation) grouping together several adjoining municipalities. The community of municipalities is managed by a community council made up of the elected representatives of the municipalities. The intermunicipality gathers 31 communes and 27000 inhabitants, which shows the relatively large size of the village in question. This political institution is mainly concerned with economic development and land use planning, but maintains close contact with several associations, notably through the cultural, social, sports, environmental and early childhood commissions.

### Data collection

Data collection began on January 1, 2022, and ended on October 1, 2022.

We identified 23 associations that frequently offer activities in the municipality. We excluded sport and art associations mainly attended by minors and those with fewer than 4 active members. Among the remaining 17, we arbitrarily selected 10 voluntary associations, two from each of the following categories: art, educational, environmental protection, sport, social. Face-to-face interaction data were collected using two methods: interviews and official documents.

We interviewed 2 members per association and 3 elected officials of the city hall, including the mayor and two deputies. We end up with 23 interviews. Participants agreed to take part in the study by signing a consent form for the collection of their personal data, in accordance with the General Data Protection Regulation (GDPR). During the interviews, members of associations were asked three topics concerning the association(s) to which they belong:

- The activities offered by the association(s), and the members who took part in these activities.

- The collaboration with other local associations.

- The interactions between the association(s) and political institutions such as the city council and the community of municipalities.

Members of the city council were asked about the composition of the committees (urbanism, culture, human service...) and the frequency of meetings. They were also asked about their contact with the community of municipalities. Regarding political institutions, we extensively use public reports of meetings where the attending persons were marked. Annual reports of associations provided clarity and were usually used as a memory basis for the interviews.

These interviews and documents provided data on 544 different activities involving 618 individuals. Since our focus is on associations and politics, we have restricted the set of

**Table 1. Distribution of memberships by category and gender.**

|  | Women(%) | Men(%) | Total Number |
|---|---|---|---|
| Art | 57.4 | 42.6 | 54 |
| Environmental-Protection | 27.0 | 73.0 | 63 |
| Human Service | 68.2 | 31.8 | 66 |
| Educational | 56.1 | 43.9 | 66 |
| Political | 38.3 | 61.7 | 227 |
| Professional | 22.2 | 77.8 | 9 |
| Sport | 42.6 | 57.4 | 68 |
| Occupational | 50.0 | 50.0 | 4 |
| Recreational | 45.5 | 54.5 | 11 |
| Total Number | 255 | 313 | 568 |

individuals to those whose membership of an association or political institution is known. This reduced the number of individuals to 474 and the number of interactions involving at least two individuals to 429. Through a snowball sampling method, we identified memberships in 54 distinct associations and 73 political institutions, which are mainly surrounding town halls.

Table 1 summarizes the composition of the sample. Associations are categorized according to their principal activity. We distinguish 8 categories of associations with corresponding examples from our case study: Human services (e.g., social work, childcare), sports clubs (e.g., running, canoe kayaking), art (e.g., theater, music, sewing), educational (e.g., association for the conservation and enhancement of cultural heritage), environmental protection (e.g., recency of species, litter pickups, fishing, vegetable gardening), recreational (e.g., organizing festive events), Occupational (e.g., inter-communal union for the management of schools, water management unions). Political members correspond to elected officials (the city council, the community of municipalities, the department...) or individuals from public institutions of an administrative nature. Lastly, we included communal maintenance agents and the secretaries of the town hall in our study because they frequently participate in organizing city tasks, meetings, and decision-making procedures in city hall. The professional category refers to these individuals. Table 1 shows the number of memberships per category. Note that, since one person can hold multiple memberships, the total number of memberships is larger than the total number of individuals.

## Hypergraph construction

The hypergraph is constructed as follows: each hyperedge corresponds to an activity, where the nodes represent the individuals participating in that activity. The frequency of the activity determines the weight of the hyperedge: (daily: 200), (weekly: 45), (monthly: 12), (quarterly: 4), (annually: 1). We denote the resulting hypergraph $\mathcal{H} = (V, E, w)$ where $V$ is the set of vertices $E$ is the set hyperedges and $w(e)$ the weight of the hyperedge $e$. For simplicity, we will use the term "edge" instead of "hyperedge". To clarify this point, here is an example of edge construction: "Alice, Tom and Peter go to the canoe club every week" results in the following edge: {Alice, Tom, Peter } with a weight of 45.

## Results

### Analysis of the hypergraph structure

In this section, we present some general features of the hypergraph $\mathcal{H}$ to better understand its structure. The hypergraph visualization using a bipartite representation is shown in Fig 1.

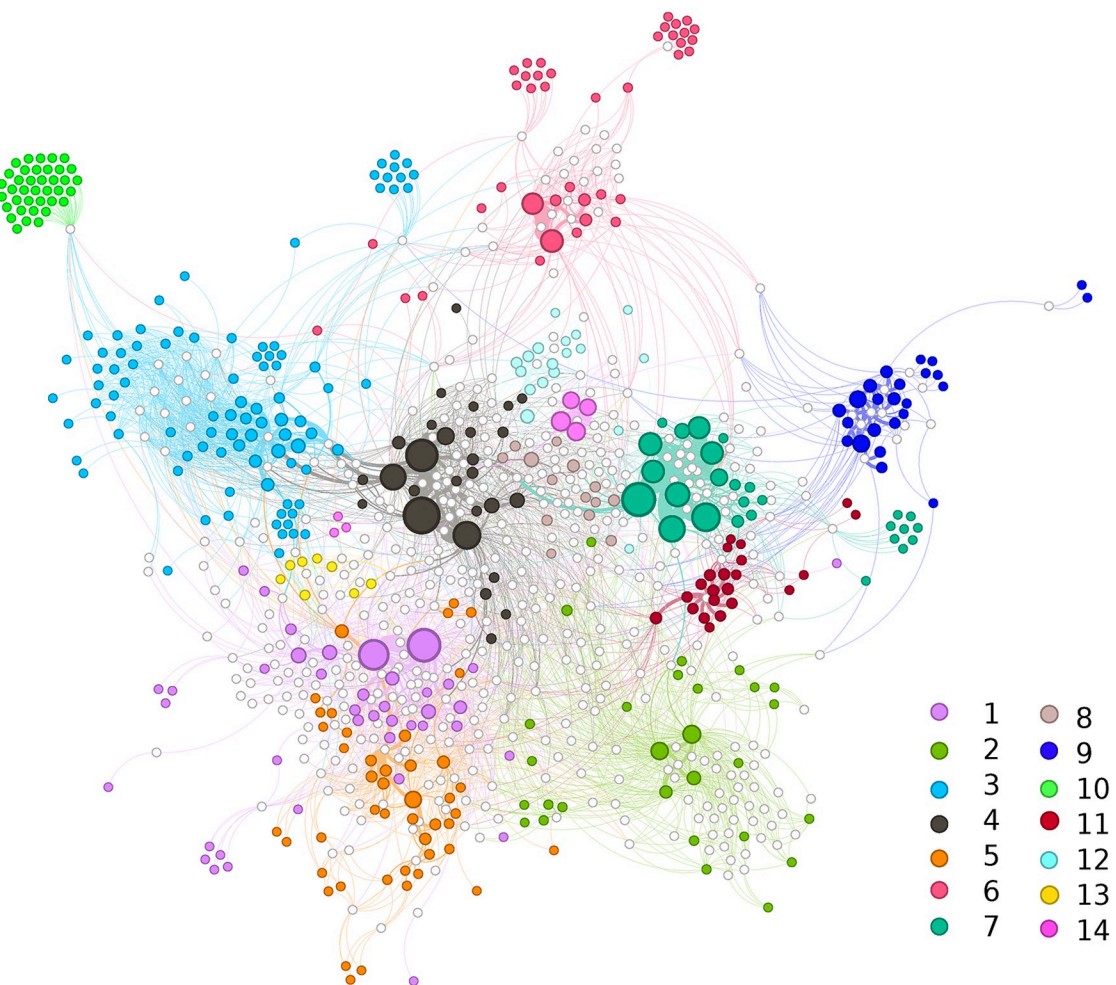

**Fig 1. Network visualization.** Wight nodes represent activities and the colored nodes represent agents. The layout is induced by the bipartite representation, where agents are connected to activities. The sizes of colored nodes are proportional to the strength of agents they represent. The width of edges connecting agents and activities are proportional to activities' weights. Colors refer to clusters obtained using the Louvain clustering algorithm with the Barber [33] modularity function. We run 700 times the clustering algorithm with random initial shuffling of nodes. The resulting partitions are very similar (see section 3 in S1 File). We select the partition with the highest modularity score. The visualization has been produced using Gephi [34].

White nodes represent activities, and colored nodes represent agents. The size of each colored node corresponds to its strength, while the color indicates the agent's cluster affiliation, as determined by a clustering algorithm that we will analyze in detail later.

Before going into a deep analysis, we start by examining some basic statistical properties of the network in Fig 2. The complementary cumulative distribution functions (CCDF) of (a) edge cardinalities, (b) edge weights, (c) node degrees, and (d) node strengths are displayed in log-log scale. There is no in-depth analysis required concerning edge weights; the CCDF is shown primarily to provide an overview of the frequency of the activities recorded in this study. We can note the total weight of the edges, which is 4391, revealing the total number of interactions recorded for this study. The cardinality of an edge corresponds to its size. As we can see, the mean cardinality of 7.22 is significantly higher than the cardinality of an edge in a standard graph, which is 2. This observation already highlights the considerable number of relatively large group interactions and justifies the use of hypergraphs.

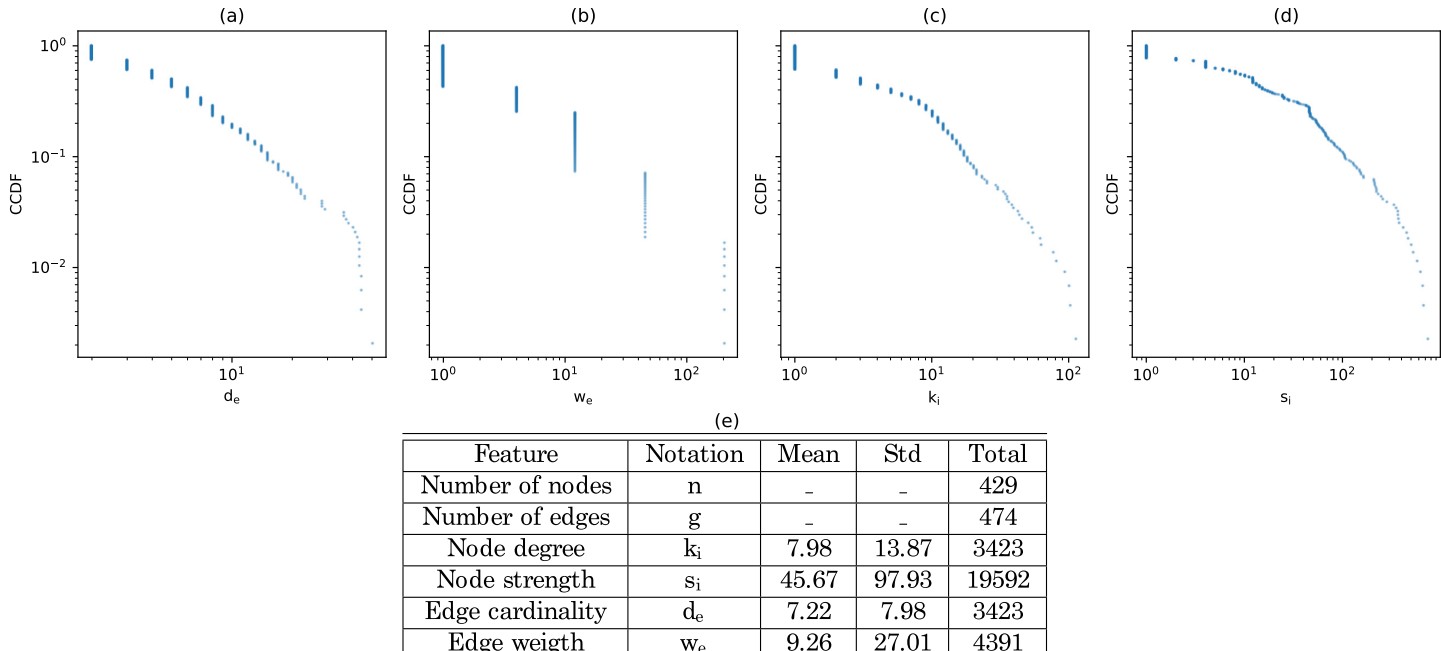

**Fig 2. Basic statistical properties of $\mathcal{H}$.** Complementary cumulative distribution functions (CCDF) of (a) edge cardinality, (b) edge weight, (c) node degree and (d) node strength in log-log scale. A summary of their statistical properties is shown in table (e).

In our case, the node degree corresponds to the variety of activities in which an individual is involved, while the node strength indicates the intensity of participation. The high standard deviation and positively skewed distribution indicate that some individuals are very active while others are more peripheral. This uneven distribution of strength can be explained in accordance with our hypothesis of community structure in two ways: either certain communities are more active, or members within communities vary in their levels of involvement.

Since we are interested in revealing cohesive groups that are densely intra-connected and loosely inter-connected, we perform a community detection algorithm based on the maximization of a quality function, namely the modularity function. Barber [33] extended the modularity function initially proposed by Newman and Givran [35] to the case of bipartite graphs (see sections 1 and 2 in S1 File for a quick review of the modularity function and the clustering algorithm).

The empirical literature lacks evidence on the efficiency of multimodal networks compared to their projections. Therefore, we perform a comparative analysis between the partitions obtained using the hypergraph and those using its clique expansion in section 3 in S1 File. We find that hypergraph clustering provides a larger number of clusters and that these clusters are more balanced in size. We argue that preserving the complex structure of connections in hypergraph clustering provides a better solution to the resolution problem associated with the modularity-based algorithm, but this assumption requires further analytical study.

The partition $\mathcal{C} = (C_1, \ldots, C_{14})$ obtained using the hypergraph clustering produces a high modularity score of 0.792. Not only does this result indicate a good partitioning by the clustering algorithm, but it also reveals the community structure underlying the hypergraph. Note that the clustering algorithm produces mixed clusters containing both agents and activities. However, for the purpose of our study, we focus on the partition of the agents, and thus, activities are removed from the clusters.

As shown in Fig 1, clusters vary in size and interaction patterns. The mean volume of the clusters is $\overline{Vol(C)} = \frac{1}{|(C)|}\sum_C\sum_{i \in C}s_i = 1399.42$ and the standard deviation of the volumes is $\sigma_{Vol(C)} = 1104.18$. Computing the Gini coefficient for the distribution of nodes strength within clusters, we find that small clusters with $Vol(C) < 1000$ have relatively low Gini coefficient (on average 0.34) indicating a more balanced distribution of node strength. In contrast, large clusters with $Vol(C) \geq 1000$ exhibit a more unbalanced distribution of node strength, with an average Gini coefficient of 0.68.

To assess whether the community structure aligns with the memberships of agents, we measure the cosine similarity between pairs of agents. For each individual $i$, we define a vector $X^i$ such that $X^i_\alpha = 1$ if agent $i$ is a member of the organization $\alpha$, and 0 otherwise. The cosine similarity between nodes $i$ and $j$ is the normalized scalar product of $X_i$ and $X_j$. Fig 3(a) shows the density distributions of *intra* and *inter* similarity. Intra-similarity is the similarity between a pair of agents within the same cluster (represented in orange), while inter-similarity (represented in blue) refers to the similarity between pairs of agents from different clusters.

The distribution of inter-similarity indicates that most of the pairs of nodes from different clusters have a vanishing similarity. This suggests that members of distinct clusters typically do not share common memberships. While the intra-similarity shows a notable proportion of strictly positive values, there is still a significant number of pairs with vanishing similarity.

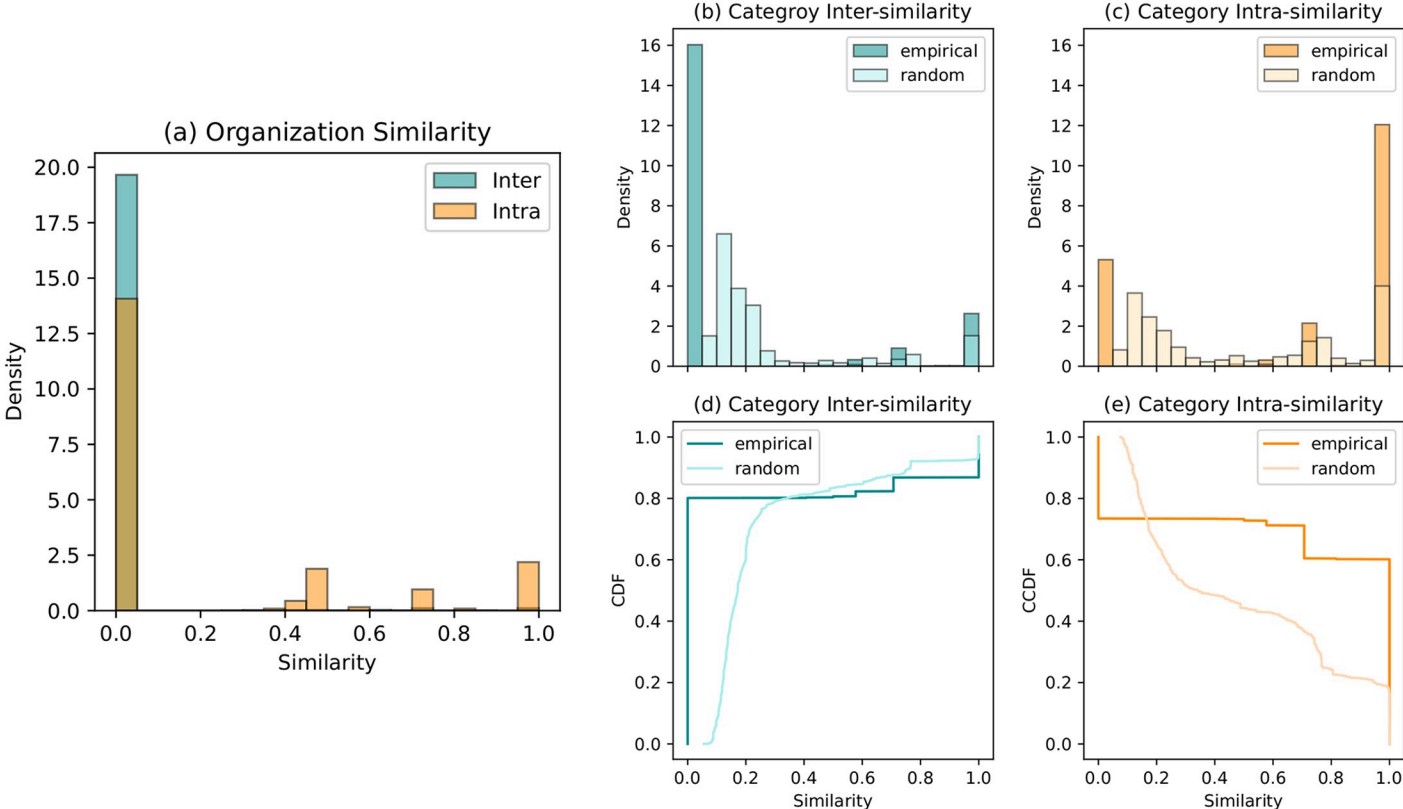

**Fig 3. Distribution of inter and intra similarity.** (a) Organization membership similarity: The intra-similarity (orange) is the similarity between pairs of agents within the same cluster, while the inter-similarity (blue) represents the similarity between pairs of agents from different clusters. (b) Category inter-similarity (c) category intra-similarity distribution and their corresponding CDF (d) and CCDF (e), respectively. The light-colored distributions represent the distribution of similarities with random assignment of organizations' categories (500 runs), while the dark-colored histograms correspond to the empirical case. The random categorization of organizations is performed while preserving the memberships and the number of organizations per category.

This implies that some nodes within the same cluster do not share any common membership. Hence, the community structure of $\mathcal{H}$ appears to be explained by organization memberships at a finer resolution, with these micro-communities merging to form larger clusters.

Now, one may wonder on what basis organizations tend to interact more or less intensively with others. Our hypothesis is that organizations of the same category may share more common activities due to their common interests. To test this hypothesis, we compute the inter- and intra-similarity, but this time taking into account organization categories. Specifically, we define the vector $Y^i$ for agent $i$ such that $Y^i_\beta = 1$ if agent $i$ belongs to at least one organization of category $\beta$, and 0 otherwise. Since this condition is less restrictive, we expect both density curves to reflect a higher probability for large similarity values. However, this behavior alone would not be sufficient to confirm our hypothesis. Therefore, we compare these empirical density curves with those obtained from random assignments of organization categories. The random categorization is performed while preserving the memberships of agents and the number of organizations per category.

The resulting curves are displayed in the right panel of Fig 3. The light colored distribution corresponds to the results of the random categorization with 500 runs, while the dark ones represent the empirical case. As expected, both empirical curves display a higher probability for large similarity values than in the previous case. As shown in Fig 3(d), an inter-similarity less than 0.35 is more likely to occur in the empirical case than in the random one. Conversely, an intra-similarity larger thant 0.16 is more probable in the empirical case than in the random one (see Fig 3(e)). The higher probability of low inter-similarity in the empirical case indicates that individuals belonging to different clusters tend not only to lack common memberships but also do not typically belong to organizations of similar categories. On the other hand, the higher frequency of high intra-similarity values in the empirical case suggests that individuals within the same cluster are more likely to be associated with organizations of similar categories, even if they are not members of the exact same organizations. This finding supports the hypothesis that organizations of the same category tend to cluster together, fostering more intense interactions among their members.

Additionally, we compare the results obtained with the hypergraph clustering to those from clustering using the clique expansion, as detailed in section 4 in S1 File. We observe that hypergraph clustering is more effective in capturing the community structure based on organization categories.

Finally, to provide a clearer understanding of the cluster compositions, Fig 4 displays the frequency of organization categories within each cluster. Each bar in the figure represents the total number of memberships by agents in a specific cluster, segmented by category. Different colors denote different categories, as indicated in the legend. The predominance of a particular color within clusters in most cases highlights the associativity of agents according to the categories of organizations they belong to.

We will not delve further into the analysis of the structure of the hypergraph. In summary, we have demonstrated that $\mathcal{H}$ exhibits a community structure. At a micro-level, communities consist of agents from the same organization, while at a broader level, they are grouped by the type of organization. These communities vary significantly in size, and large clusters include a core of highly active individuals surrounded by less active members.

## Centralities

With a general understanding of the interactions between members of associations and political institutions, we now move on to a more detailed analysis to address the following question: What types of behavior facilitate individuals' engagement with local politics? To explore this,

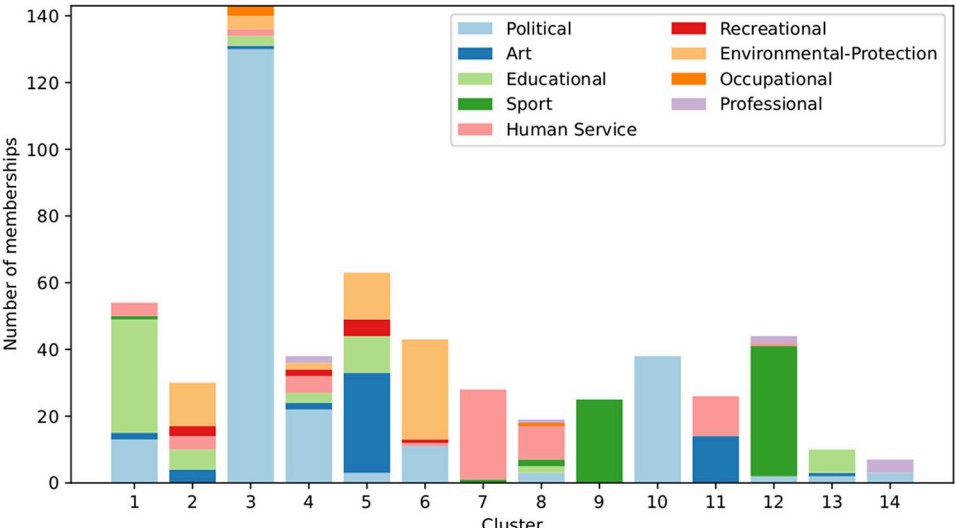

**Fig 4. Cluster composition.** Each bar represents a cluster, with the x-axis indicating different clusters. The height of each bar corresponds to the total number of memberships within the cluster. Each unit rectangle within a bar represents one membership of an agent, with individuals who are members of multiple categories contributing one rectangle per category. The color of each unit rectangle reflects the organization category associated with the membership.

we introduce and revisit several centrality measures. While it is possible to normalize all centrality vectors by setting their $l_2$-norm to one, this step is unnecessary for our purposes, as we are primarily interested in computing correlations in the next section.

**Edge-dependent degree centralities.**

- *Strength*: First, we recall the strength centrality

$$s_i = \sum_{e:i\in e} w_e \tag{1}$$

This measure reflects the intensity of an individual's participation across all types of activities organized by associations and political institutions.

- *Political participation*: We also need to define a measure to quantify access to local politics, which we call *political participation*, denoted as $p_i$ for agent $i$. First we define a set of agents that we call the *political body* and note it $\mathcal{P}$. An agent belongs to $\mathcal{P}$ if he or she is an elected official member of a political institution. Let $e$ an edge corresponding to an activity proposed by an organization, we make the following hypothesis: if the proportion of individuals from the political body in $e$ is large, then it is more probable that the activity represented by $e$ is a 'political activity', i.e., contributes to making political decisions. Conversely, if this proportion is very low, then it is more probable that the activity is 'apolitical'. This leads to the following expression:

$$p_i = \sum_{e:i\in e} \frac{|e \cap \mathcal{P}|}{d_e} w_e \tag{2}$$

where $|e \cap \mathcal{P}|$ is the number of agents in $e$ that are from the political body. This measure can be seen as an extension of the usual degree centrality, becoming an edge-dependent degree centrality.

- *Diversity coefficient*: We introduce another edge-dependent degree centrality, which measures an agent's ability to participate in 'heterogeneous' activities involving individuals from different communities. The heterogeneity of an activity represented by an edge $e$ is captured by the Shannon entropy:

$$h(e) = - \sum_{C_j \in \mathcal{C}} \frac{|e \cap C_j|}{d_e} \log\left(\frac{|e \cap C_j|}{d_e}\right)$$

where $\frac{|e \cap C_j|}{d_e}$ is the proportion of individuals in $e$ that are in cluster $C_j$. A high entropy corresponds to an activity $e$ that gathers individuals from various organizations. For a given agent $i$, the corresponding edge-dependent degree centrality is denoted $o_i$ and has the following expression:

$$o_i = \sum_{e:i \in e} h(e) w_e \tag{3}$$

**Eigenvector centralities.**   We recall the centrality measures proposed by Tudisco and Higham [36]. They define a general framework for computing node and edge eigenvector centrality for hypergraphs where the importance $y_e$ of an edge $e$ is a non-negative number proportional to a function of the importances of the nodes in $e$, and similarly, the importance $x_i$ of a node $i$ is a non-negative number proportional to a function of the importances of the edges it participates in. The centrality measures are defined as follows:

$$x_i \propto g\left(\sum_{e:i \in e} w_e f(y_e)\right), \quad y_e \propto \psi\left(\sum_{i \in e} \varphi(x_i)\right)$$

where $f, g, \psi, \phi$ are four functions, possibly non-linear, that must be specified by the user.

- *EV linear*: The choice of the functions $f = g = \psi = \phi(x) = id$ essentially corresponds to the standard eigenvector centrality applied to the clique expansion.
- *EV log-exp*: The log-exp eigenvector centrality corresponds to the choices $f = id$, $\phi(x) = \ln(x)$, $\psi(x) = \exp(x)$ and $g(x) = \sqrt{x}$. This choice corresponds to a nonuniform hypergraph version of the tensor Z-eigenvector centrality for uniform hypergraphs [36]
- *EV max*: In this case, we set $\phi = \psi = id$, $f(x) = x^\alpha$ and $g(x) = x^{1/\alpha}$ with $\alpha = 10$. Based on the fact that the function $\left(\sum_{i=1}^n \lambda_i^\alpha\right)^{1/\alpha} \xrightarrow{\alpha \to \infty} \max_i(\lambda_i)$, we have $x_i \approx \max\{y_e, e: i \in e\}$. Hence, a node is central if it is part of at least one important edge.

**Cluster z-score.**   Let $i$ be a node and $C$ be the cluster to which $i$ belongs. We define the cluster z-score centrality as the relative strength of $i$ in $C$:

$$z_i = \frac{s_i - \overline{s_C}}{\sigma_C}$$

where $\overline{s_C}$ is the mean strength of individuals present in $C$ and $\sigma_C$ is the standard deviation of the strengths of individuals in $C$.

**Betweenness.**   Betweenness centrality for a given node $i$ is defined as the proportion of shortest paths between all pairs of nodes that pass through $i$. Specifically, it is calculated as the

**Table 2. Pearson correlation coefficients between political participation and various centrality measures for two distinct groups: Political body $\mathcal{P}$ and association members $V \setminus \mathcal{P}$.**

| Centrality measure | $\mathcal{P}$ | $V \setminus \mathcal{P}$ |
|---|---|---|
| Strength | 0.990 | 0.572 |
| Diversity | 0.800 | 0.810 |
| EV linear | 0.800 | 0.139 |
| EV log exp | 0.884 | 0.185 |
| EV max | 0.074 | 0.402 |
| Cluster z-score | 0.687 | 0.512 |
| Betweenness | 0.437 | 0.585 |
| Closeness | 0.099 | 0.134 |

number of shortest paths that include node $i$, divided by the total number of shortest paths in the network. For this study, we compute betweenness centrality using the clique expansion of $\mathcal{H}$

**Closeness.** Closeness centrality for a node $i$ is the reciprocal of the sum of the shortest path distances from node $i$ to all other nodes in the network. It reflects how close a node is to all other nodes. We compute closeness centrality using the clique expansion of $\mathcal{H}$.

## Correlation with political participation

Having defined the political participation rate, and various centrality measures, in this section, we can now address the question posed in the previous section: What types of behavior facilitate individuals' engagement with local politics?To this end, we compute the Pearson correlation between $p_i$ and centrality measures. The results are shown in Table 2. The Pearson correlation coefficients are computed for two distinct sets of nodes: the political body $\mathcal{P}$, and the set of individuals who are exclusively members of associations $V \setminus \mathcal{P}$.

For the set of political figures $\mathcal{P}$, most centrality measures show a very high Pearson correlation with political participation, except for EV max, betweenness, and closeness centralities. Notably, strength centrality stands out as a particularly strong predictor, with a correlation of 0.990, indicating that higher overall activity levels are closely associated with higher political participation. This correlation is higher than that between the diversity coefficient and political participation (0.800), suggesting that for political figures, overall activity is a better indicator of political participation than the diversity of those activities.

For members of $V \setminus \mathcal{P}$, the diversity coefficient shows the highest correlation with political participation (0.810) compared to other measures, including strength centrality (0.572). This indicates that, for association members, engaging in activities that involve people from various communities is more strongly associated with political participation. Additionally, strength centrality and cluster z-score centrality both demonstrate significant correlations with political participation. This indicates that, people who are active, in general, and compared to others in their community, have greater access to local politics. Betweenness centrality also has a high correlation, reflecting the importance of nodes that connect other nodes and potentially bridge communities. This is related to the concept captured by the diversity coefficient, which also involves bridging between communities. We also note that the correlation with EV max is moderate (0.402), indicating some predictive power for non-political members. Conversely, the correlation with EV linear is much lower (0.139). This suggests that traditional eigenvector centrality does not capture the higher-order interactions in this context effectively. This highlights the need for more nuanced analytical approaches. Finally, closeness centrality shows no

significant correlation (0.134) with political participation, indicating that proximity alone does not explain political engagement for association members. Similarly, the correlation with EV log-exp is very low.

## Discussion

In this section, we synthesize and discuss the results from the previous sections qualitatively. First, we analyze the position of the political body within the network. Fig 4 shows that agents in $\mathcal{P}$ are mainly distributed across five clusters (1, 3, 4, 6 and 10). Notably, the three largest clusters (3, 4, and 10) are almost exclusively composed of politicians. Not surprisingly, members of the political body have, on average, a political participation rate that is eight times higher than that of association members. In addition, in Table 2, the fact the strength centrality correlates better with political participation than the diversity coefficient indicates that participating in political activities and in homogenous activities are mutually reinforcing. All these observations show that the political body forms relatively exclusive communities where the activities, mainly political, occur in a political "entre-soi." It is important to recall that this study focuses on associations and the city council within a village. Many members of the political body are not elected officials of the town hall but come, through a snowball, from higher authorities such as the community of municipalities. Thus, the way the data was collected suggests that, from the perspective of the members of the associations, these political communities seem relatively closed. Engaging with them and participating in political activities seems to be reserved to some extent for certain privileged members. This inequality of access to local politics for association members is reflected in the Gini coefficient for political participation of 0.785, close to 1.

Table 2 demonstrates that association members who participate in activities involving individuals from diverse communities tend to have greater access to local politics. While one might assume that political activities are inherently heterogeneous and that the diversity coefficient is merely an artifact, the Pearson correlation between the heterogeneity of an activity and its politicization is 0.139, indicating a positive but not fully conclusive relationship.

Furthermore, by comparing the results obtained for strength centrality, we infer that an individual's general social participation is less predictive of their political participation than their involvement in heterogeneous activities. This conclusion is further supported by the high level of betweenness centrality for the clique expansion of the hypergraph. Indeed, these heterogeneous activities are all the more important since the hypergraph has a community structure. Hence, they serve as bridges between different organizations, providing participants with visibility in the public sphere and potentially granting them legitimacy in the political arena.

Until now, this analysis has focused on interaction data to explain the access to politics for association members. We will now examine the status of association members. Table 3 shows the point biserial correlation between holding the position of association president and various centrality measures.

The results suggest that leadership roles, such as being a president of an association, are strongly associated with proximity to local politics (0.384) and participation in heterogeneous and bridging activities, as reflected by the high correlations with diversity (0.397) and betweenness (0.369) centralities. The EV max centrality shows a moderate positive correlation (0.298, p = 0.000) with being a president, indicating that participating in at least one large activity has some predictive power regarding leadership roles, though it is not as strong as diversity or betweenness. While traditional measures like strength (0.243) also play a role, eigenvector-based centralities, especially EV linear and EV log exp, do not effectively capture the characteristics of individuals who assume leadership positions. Additionally, the difference in

**Table 3. Point biserial correlation between being president of an association and centrality measures.**

| Centrality | Point biserial | p-value |
|---|---|---|
| Political participation | 0.384 | 0.000 |
| Strength | 0.243 | 0.000 |
| Diversity | 0.397 | 0.000 |
| EV linear | 0.036 | 0.571 |
| EV log expo | 0.005 | 0.941 |
| EV max | 0.298 | 0.000 |
| Cluster z-score | 0.312 | 0.000 |
| Betweenness | 0.369 | 0.000 |
| Closeness | 0.134 | 0.034 |

correlation between cluster z-score (0.312) and strength suggests that these leaders are more active within their own organization than outside them.

Overall, the network exhibits a community structure closely aligned with organizational affiliation. The distribution of strengths within these communities is unevenly distributed. At the head of these communities are the presidents of the associations. In addition to being very active in their associations, the presidents are also the gateways to other organizations and participate in activities that bring together people from different communities. Crucially, it is not merely their strong integration within their own communities that defines their leadership roles. Instead, their visibility in the public sphere and their capacity to bridge different organizations emerge as pivotal characteristics underpinning their political legitimacy. This analysis points out that the associations' modes of governance involving a president assigned to the role of the representation of the association, often exacerbates power imbalances rather than equalizing them among members. In many cases, as observed in our study, presidents hold a significant number of mandates, sometimes being founders of the organization, and are regularly re-elected during annual general assemblies. Studies conducted in 2011 and 2014 in France by the Ministère de la Ville de la Jeunesse et des Sports [37] identify a governance model known as "tightened governance," which aptly describes our case. "In this type of association, governance is embodied in one or more omnipresent and charismatic persons: the president and/or the leader [. . .] the president does not leave much room for other internal stakeholders who tend to rely on his dynamism." They point out that this concentration of power often leads to difficulties in leadership renewal, probably because other members do not feel as much integrated in the collective project and do not feel legitimate to run for the office. To address the challenges associated with this governance model, some civic associations have proposed solutions. For instance, creating intermediate collegial bodies to feed back ideas to the decision-making body. This is intended to involve a variety of members and diversify the places of expression and decision. Other organizations fix the number of mandates such that the renewal of the leaders become mandatory.

## Conclusions

In this paper, we examined social interactions between members of associations and local political institutions in a French city with nearly two thousand residents. We modeled these group interactions using a hypergraph. We used a community detection algorithm specifically designed for hypergraphs to reveal the architecture of the connections between individuals. Our analysis revealed that the hypergraph exhibits a community structure rooted in the organizations involved in our study at a micro-level, with these organizations clustering together

by category to form larger communities. This confirms hypothesis 1 in the Related work and hypotheses section. These communities differ in size and in the way they are connected to each other. The fat tail of the distribution of strength among the nodes in the global network is still observed within the clusters.

We define an edge-dependent degree centrality called *political participation*, which quantifies access to local politics. To explain people disparity in political participation, we introduce another edge-dependent degree centrality, the *diversity coefficient*, which measures the ability to participate in activities involving people from different communities. We compare the correlations between political participation and multiple centralities, including the one we define and others from the literature. We find that involvement in heterogeneous activities is the key element to explaining political participation among members of voluntary associations. This confirms hypothesis 2 in the Related work and hypotheses, and in a more general context, aligns with Granovetter's theory of the strength of weak ties [19] and the Burt's theory of structural holes [20]. When we examine the status of association members, we observe that association presidents are most likely to participate in such activities and have the greatest access to local politics.

The purpose of this study is not to determine a causal relationship between people's interactional behavior and their political participation [18, 38]. One may argue that the diplomatic trait of individuals is the one that allows them to participate in activities that gather individuals from different backgrounds and is a key characteristic to become a president of an association and gain some legitimacy in the political space. On the other hand, as being the president of an association, one should represent the association during formal meetings with political institutions and association partners. This role makes them the spokesperson of associations and forces them to position themselves as intercommunity bridges through their participation in heterogeneous activities. Nonetheless, these results highlight the inequalities inherent in associations' governance structures, which should be further examined and addressed.

Our survey does not allow us to determine whether today's association members will be tomorrow's local elected representatives. However, we note that within the current municipality studied, several members of the municipal council have been, or still are, involved in local associations. This observation aligns with the findings of the Nevers and Bages survey [39], which compares 436 towns with fewer than 2500 inhabitants in the Midi-Pyrénées region and found an undeniable correlation between mayors' past involvement in associations and their subsequent rise to the presidency of the municipality. Similarly, Malet's study, which compares more than 50 communes of 500 to 800 inhabitants in 40 different French departments, indicates that "80% of association leaders in these communes have a relationship with their municipality" [40]. These findings suggest that, beyond the professional sphere, political activity can be understood as a social activity where associations serve both as training grounds for developing political techniques and as potential springboards toward the professionalization of politics.

While our study emphasizes the role of social ties, we acknowledge that external forces—such as media coverage, public demonstrations, or political campaigns—can also play a crucial role in shaping political behavior. It's important to note that this study is based on correlation rather than regression analysis, which is more commonly used in surveys. Although it would be valuable to comparatively assess the impact of social ties versus other external forces, doing so would require a different approach to data collection, such as a broader survey. Our study, which involves in-depth interviews with only 23 individuals, is limited in its scope. Additionally, our focus on organizational membership ties means we did not account for other types of social connections, such as those formed through children's school activities, workplaces, or community spaces like cafés. A more comprehensive study would require surveying a larger population to capture these diverse social ties.

Of course, the location of our study in a single municipality in France makes the generalization of our results questionable. We think that the connections between people would have a different pattern in larger urban cities, probably with a larger number of associations building up more clustered communities connected with fewer bridges, and where only a reduced number of presidents would be in contact with elected officials. Additionally, in cultural contexts where other governance models for associations are prominent, different conclusions would likely arise. However, this limitation is almost inherent to our sociometric techniques, as conducting such a detailed study on a national scale would be expensive. We believe that the goal of such a study is to provide a more accurate understanding of the mechanisms that lead to unequal representativeness of association members in the political arena.

Nevertheless, our work is generalizable in light of other studies conducted in different contexts [6, 18]. While these studies consider social diversity based on religion, ethno-cultural background, and political orientations, we define social network diversity as the ability to interact with distinct communities. Regardless of its definition, social heterogeneity consistently correlates with increased political participation.

Finally, we would like to say a few words about the method used in this study, which probably makes up the originality of this work. We drew on methods often used in qualitative sociology and anthropology, by asking fairly open-ended questions during interviews rather than detailed, pre-formulated questionnaires. This approach allowed us to adapt our inquiries to the specific characteristics of each organization, such as the size of the association or its affiliation with a national federation. Our data collection relied heavily on documents provided by respondents during interviews, including annual reports, agendas, and photographs. The data collected were then analyzed quantitatively with macro-level network techniques. We choose a hypergraph modeling rather than a traditional graph because the structure of the data, in which people participate in activities, is better suited to hypergraph representation. This approach allowed us to preserve all the information without projecting the hypergraph into its clique expansion. To advocate for preserving higher order interactions, we compare the performance of the community detection algorithm for hypergraph and its clique expansion in section 3 in S1 File. We argue that preserving the complex structure of interactions in hypergraph clustering better addresses the resolution problem associated with the modularity-based algorithm and permits a better recovery of the organizational categories. This empirical observation should spruce an in-depth analytical analysis of this issue.

Another advantage of preserving the hypergraph structure is the opening of a wider range of centrality measures. By considering node centrality relative to the centrality of the edges it belongs to, we can develop edge-dependent degree centralities that leverage the composition of edges. Similarly, defining node-dependent centralities for edges allows for various extensions of eigenvector centrality. Our findings indicate that the eigenvector centrality in its linear form (applied to the clique expansion of the hypergraph) does not correlate with political participation. However, when introducing non-linearity through the eigenvector max centrality, which leverages the more complex structure of the hypergraph and accounts for edge cardinality, we observe a better correlation with political participation. While this last centrality is not intended to mimic social behavior, we hope that this result encourages going further in this work and investigates other sets of functions and their corresponding social notions.

## Supporting information

**S1 File. Supplementary methods.**
(PDF)

## Acknowledgments

Our special thanks go to Antoine Mandel for his invaluable comments and advice throughout this study. We also extend our gratitude to Camille Roth and Alain Barrat for their insightful feedback and guidance.

## Author Contributions

**Conceptualization:** Amina Azaiez.

**Data curation:** Amina Azaiez.

**Formal analysis:** Amina Azaiez.

**Investigation:** Amina Azaiez, Robin Salot.

**Methodology:** Amina Azaiez.

**Project administration:** Amina Azaiez.

**Software:** Amina Azaiez.

**Supervision:** Amina Azaiez.

**Validation:** Amina Azaiez.

**Visualization:** Amina Azaiez.

**Writing – original draft:** Amina Azaiez.

**Writing – review & editing:** Robin Salot.

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
