## [Decision Letter · Decision Letter 0]

26 Jun 2024

PONE-D-23-42942Political Participation and Voluntary Associations : A Hypergraph Case StudyPLOS ONE

Dear Dr. Azaiez,

Thank you for submitting your manuscript to PLOS ONE. After careful consideration, we feel that it has merit but does not fully meet PLOS ONE’s publication criteria as it currently stands. Therefore, we invite you to submit a revised version of the manuscript that addresses the points raised during the review process.

We look forward to receiving your revised manuscript.

Kind regards,

Hocine Cherifi

Academic Editor

PLOS ONE

Reviewers' comments:

Reviewer's Responses to Questions

**Comments to the Author**

1. Is the manuscript technically sound, and do the data support the conclusions?

Reviewer #1: Yes

Reviewer #2: Yes

2. Has the statistical analysis been performed appropriately and rigorously? 

Reviewer #1: Yes

Reviewer #2: Yes

3. Have the authors made all data underlying the findings in their manuscript fully available?

Reviewer #1: Yes

Reviewer #2: Yes

4. Is the manuscript presented in an intelligible fashion and written in standard English?

Reviewer #1: Yes

Reviewer #2: Yes

5. Review Comments to the Author

Reviewer #1: The article brings in its essence a proposal that brings originality and relevance to the field. The summary is well structured and adequately presents the article and its findings. The observations to be made, in this opinion, seek to improve the reader's experience, given the relevance of the work, as well as explore the study's potential contribution to the field of understanding political phenomena, especially participation, in democratic societies.

In the introduction, the context is clear, and the objectives of the article are presented precisely.

From a writing point of view, a small revision in long and exhaustive paragraphs is recommended, especially in the presentation of theories in section 2. In this way, it would be possible to improve readers' experience in relation to the study, understanding the origin of the study variables, and more clearly, when analyzing, the relationship with theory.

Furthermore, in section 2, it would be possible to make the concept of political participation clearer, given its complexity and the relationship with other concepts, which derive from its understanding.

The methodology is relevant, it is clearly exposed, as is the data analysis process. As pointed out in the article, this is the most relevant point of the work, considering that the approach has a strong relationship with practical work in the field of politics, at the same time that it advances the understanding of the creation of ties in the process of political participation, observing both citizens, from multiple sectors, as well as institutions and their roles.

As for the conclusions of the study, although very well presented and structured, it would be appropriate, as a suggestion, for a slightly greater discussion of what was observed, not only as political participation in this specific context, but also to explore some possibilities for understanding the democratic process as observed. This can inspire paths for new works and approaches in different contexts.

Reviewer #2: The theoretical contributions of the article are clear and significant. The study extends existing theories of social networks and political participation by applying them in a novel context using advanced methodological tools. The use of hypergraph analysis to capture complex, higher-order interactions in social networks is particularly innovative.

I must commend the authors for revisiting well-grounded theories, particularly Granovetter's strength of weak ties and Burt's structural holes concepts, a practice that more political scientists should adopt. I also think that the attempt to step away from self-reported data and shift to real-world data from a single municipality has the potential to capture higher-order interactions that survey designs and even traditional network analysis research may miss. The authors provide convincing evidence, particularly regarding the contribution of heterogeneous activities’ participation on political participation for political participation among association members.

That being said, I have a few misgivings and suggestions which I believe the authors could address in a revised version of the paper.

1. What was the selection criteria for the 2 organizations out of each association group?

2. Why external motivation must be limited to neighbors and not include other external forces, such as those I see active on the news (a farmers road blockade, someone running for mayor, etc.), which I am not familiar with, but push me to do something.

3. The study is limited to a single municipality in France, which with its higher than average retirees and farming, hampers generalizations, even within the French population. It is not just how it differs in nature to Paris, , but also demographically. I think the authors can do better job in explaining that the nature of the individuals in town are not as important as their findings.

4. I understand that the explanation is driven by organizational membership ties, however, as someone who lives in the village of 3,000 people, many of the social ties within our community are based on the social connections originally created by classroom membership of our children at school, not through our vocation, art or sport or social groups. Till this day, years after my kids left home, it is the parents of my kids’ peers which compose my ties within the community, the ones who inform me and activate me, and vice versa. Obviously, this may be different for the older residents of an agricultural rural community, but then we are back to generalizability.

5. Given PLOS One’s general readership, you may want to provide benefit from a more detailed explanation of the hypergraph methodology and its advantages over traditional network analysis approaches. An illustration may do wonders here.

6. The authors find that “people who are active, in general, and compared to others in their community, have more access to local politics”. However, given the correctional nature of this study, could it be that those who are active in one dimension (politics) become active in other dimensions (art, social, etc.), or perhaps both are driven by an external force (extroversion, for example). Is there a sub sample within your data which at the beginning of the year did not participate in politics but were drawn to it later by their ties?

7. The authors may want to check the following. The first one for its method and the second as another example of the benefits of network heterogeneity.

Like-minded and cross-cutting talk, network characteristics, and political participation online and offline: A panel study. Jörg Matthes, Franziska Marquart, Christian von Sikorski. Communications 46 (1), 113-126, 2021.

Scheufele, D. A., Hardy, B. W., Brossard, D., Waismel-Manor, I. S., & Nisbet, E. (2006). Democracy Based on Difference: Examining the Links Between Structural Heterogeneity, Heterogeneity of Discussion Networks, and Democratic Citizenship. Journal of Communication, 56(4), 728–753.

Quitellier et al in text introduction is missing parentheses.

6. PLOS authors have the option to publish the peer review history of their article (what does this mean?). If published, this will include your full peer review and any attached files.

Reviewer #1: No

Reviewer #2: No

---

## [Author Response · Author response to Decision Letter 0]

19 Sep 2024

1. Please ensure that your manuscript meets PLOS ONE's style requirements, including

those for file naming. The PLOS ONE style templates can be found at

https://journals.plos.org/plosone/s/file?id=wjVg/PLOSOne_formatting_sample_main_body.pd

f and

https://journals.plos.org/plosone/s/file?id=ba62/PLOSOne_formatting_sample_title_authors_

affiliations.pdf

2. Please update your submission to use the PLOS LaTeX template. The template and more

information on our requirements for LaTeX submissions can be found at

http://journals.plos.org/plosone/s/latex.

Thank you for the guidance. We have updated our submission to use the PLOS LaTeX

template

3. When completing the data availability statement of the submission form, you indicated that

you will make your data available on acceptance. We strongly recommend all authors decide

on a data sharing plan before acceptance, as the process can be lengthy and hold up

publication timelines. Please note that, though access restrictions are acceptable now, your

entire data will need to be made freely accessible if your manuscript is accepted for

publication. This policy applies to all data except where public deposition would breach

compliance with the protocol approved by your research ethics board. If you are unable to

adhere to our open data policy, please kindly revise your statement to explain your

reasoning and we will seek the editor's input on an exemption. Please be assured that, once

you have provided your new statement, the assessment of your exemption will not hold up

the peer review process.

Thank you for your guidance. In response, we have made the data useful for our study

available in a GitHub directory. The URL for accessing this data is specified in the Data

Review URL section of our submission. We hope this meets the requirements for data

sharing and facilitates the review process.

4. Please review your reference list to ensure that it is complete and correct. If you have

cited papers that have been retracted, please include the rationale for doing so in the

manuscript text, or remove these references and replace them with relevant current

references. Any changes to the reference list should be mentioned in the rebuttal letter that

accompanies your revised manuscript. If you need to cite a retracted article, indicate the

article’s retracted status in the References list and also include a citation and full reference

for the retraction notice.

Thank you for the reminder. We have thoroughly reviewed our reference list to ensure its

completeness and accuracy. In response to the reviewer comments, we have added the

following reference:6. Scheufele DA, Hardy BW, Brossard D, Waismel-Manor IS, Nisbet E. Democracy Based on

Difference: Examining the Links Between Structural Heterogeneity, Heterogeneity of

Discussion Networks, and Democratic Citizenship. Journal of Communication.

2006;56(4):728–753. doi:10.1111/j.1460-2466.2006.00317.x

10. McClurg SD. Social Networks and Political Participation: The Role of Social Interaction in

Explaining Political Participation. Political Research Quarterly. 2003;56(4):449–464.

doi:10.2307/3219806.

11. Koebel M. Les profits politiques de l’engagement associatif. Regards Sociologiques.

2000;20:165–176

12. Hamidi C. ´El ´ements pour une approche interactionniste de la politisation: Engagement

associatif et rapport au politique dans des associations locales issues de l’immigration.

Revue fran ¸caise de science politique. 2006;56(1):5. doi:10.3917/rfsp.561.0005

13. Rougier C. Tenir le politique `a distance : une capacit ´e citoyenne ? Retour sur l’ ´echec

d’une tentative d’instrumentalisation partisane d’une association de loisirs. Recherches

sociologiques et anthropologiques. 2015;46(1):67–88. doi:10.4000/rsa.1388.

14. Sim ´eant J. Un humanitaire “ apolitique ” ? D ´emarcations, socialisations au politique et

espaces de la r ´ealisation de soi. In: Belin, editor. La politisation; 2003.Available from:

https://hal.science/hal-03872136.

21. Ohlemacher T. Bridging People and Protest: Social Relays of Protest Groups against

Low-Flying Military Jets in West Germany. Social Problems. 1996;43(2):197–218.

doi:10.2307/3096998.

24. Mutz DC. The Consequences of Cross-Cutting Networks for Political Participation.

American Journal of Political Science. 2002;46(4):838. doi:10.2307/3088437

25. Battiston F, Cencetti G, Iacopini I, Latora V, Lucas M, Patania A, et al. Networks beyond

pairwise interactions: Structure and dynamics. Physics Reports. 2020;874:1–92.

doi:10.1016/j.physrep.2020.05.00

27. Ugander J, Backstrom L, Marlow C, Kleinberg J. Structural diversity in social contagion.

Proceedings of the National Academy of Sciences. 2012;109(16):5962–5966.

doi:10.1073/pnas.1116502109

28. Mønsted B, Sapie ˙zy ´nski P, Ferrara E, Lehmann S. Evidence of complex contagion of

information in social media: An experiment using Twitter bots. PLOS ONE.

2017;12(9):e0184148. doi:10.1371/journal.pone.0184148.

29. Iacopini I, Petri G, Barrat A, Latora V. Simplicial models of social contagion. Nature

Communications. 2019;10(1):2485. doi:10.1038/s41467-019-10431-6.

30. Sahasrabuddhe R, Neuh ¨auser L, Lambiotte R. Modelling non-linear consensus

dynamics on hypergraphs. Journal of Physics: Complexity. 2021;2(2):025006.

doi:10.1088/2632-072X/abcea3

32. Bastian M, Heymann S, Jacomy M. Gephi: An Open Source Software for Exploring and

Manipulating Networks; 2009. Available from:

http://www.aaai.org/ocs/index.php/ICWSM/09/paper/view/1

36. Matthes J, Marquart F, Sikorski CV. Like-minded and cross-cutting talk, network

characteristics, and political participation online and offline: A panel study. Communications.

2021;46(1):113–126. doi:10.1515/commun-2020-2080

37. Nevers JY, Bages R. Les maires des petites communes face aux enjeux de la

diversification du monde rural, une enquˆete aupr`es de 436 ´elus de Midi-Pyr ´en ´ees; 2008.

Available from: https://shs.hal.science/halshs-00216201.

38. Malet J. Les associations, source de vitalit ´e du milieu rural ?:. Pour.

2009;N°201(2):97–102. doi:10.3917/pour.201.0097.

Reviewer #1: The article brings in its essence a proposal that brings originality and

relevance to the field. The summary is well structured and adequately presents the article

and its findings. The observations to be made, in this opinion, seek to improve the reader's

experience, given the relevance of the work, as well as explore the study's potential

contribution to the field of understanding political phenomena, especially participation, in

democratic societies.

In the introduction, the context is clear, and the objectives of the article are presented

precisely.

From a writing point of view, a small revision in long and exhaustive paragraphs is

recommended, especially in the presentation of theories in section 2. In this way, it would be

possible to improve readers' experience in relation to the study, understanding the origin of

the study variables, and more clearly, when analyzing, the relationship with theory.

-

Thank you for your positive feedback and constructive suggestions. In response, we

have condensed several lengthy paragraphs in Section 2 to reduce redundancy and

emphasize key results from the literature. These revisions are highlighted in blue in

the marked-up version.

Furthermore, in section 2, it would be possible to make the concept of political participation

clearer, given its complexity and the relationship with other concepts, which derive from its

understanding.

-

Thank you for highlighting this important point. We have provided a clearer definition

of political participation in the Introduction section:

Indeed, this literature usually limits the study of “political participation” to

voting patterns and participation rates in national elections. However, social participation and

political participation can be seen as a two-way flow. The concept of “politics” can be

extended to all activities that serve to organize social groups, without being limited to the act

of voting. In this sense, the boundaries of politics can be expanded by taking it out of its

purely professional field (Koebel 2000; Hamidi 2006; Rougier 2015). From this perspective, it

is not necessary tohave an established position in political spaces to effectively mobilize

political techniques. For example, techniques such as the ability to speak in public, to

distribute individual tasks with authority or horizontality for the organization of collective

activities, to establish strategies for seizing power or contesting the authorities instituted by

democratic tools, etc., can be found in both the associative and political spheres. Beyond

these technical similarities, these associations can also influence politicians at the city level

through their frequent contacts with city council members, for example, when organizing

social events or setting up social services such as childcare or help for the elderly. In

addition, politicians may see these associations as potential channels for contacting and

convincing their members, thereby becoming vehicles for electoral legitimization (Koebel

2000). Without going so far as to homogenize the two spheres (Sim ´eant 2003), we define

political participation as access to local politics. More formally, we measure the frequency

with which people participate in activities involving local elected officials.

-

We also have explained the relationship between political participation and the

importance of bridging ties in subsection Related work and hypotheses:

Applying this theoretical concept to explain the level of political participation, bridging ties

should enhance political knowledge and understanding, thereby increasing participation in

political activities (Crenson 1978; Ohlemacher 1996; Eveland and Kleinman 2013). The

results of Crenson’s 1978 (1978) study in Baltimore seem to support this theory. He

compared community associations in six neighborhoods to reveal the factors behind the

associations’ successful operation. He found that residents with loose-knit neighborhood

friendship ties were more likely tobe informed about community associations and their

associations better met their interest. Another comparative study by Ohlemacher

(Ohlemacher 1996) reached similar results. His analysis of the organization membership

affiliations networks of two protest groups in West Germany shows that successful

mobilization correlates with the presence of structural bridging links.

Moreover, Ohlemacher insists on the need for these links, as their absence in the weak

mobilization group leads to network fragmentation. This result is also highlighted by Eveland

and Kleinman (2013) in their study of 25 voluntary groups in a university. Comparing their

general and political discussion networks, they found that political discussion networks are

more likely to be broken down into subcomponents than general discussion networks.

The methodology is relevant, it is clearly exposed, as is the data analysis process. As

pointed out in the article, this is the most relevant point of the work, considering that the

approach has a strong relationship with practical work in the field of politics, at the same time

that it advances the understanding of the creation of ties in the process of political

participation, observing both citizens, from multiple sectors, as well as institutions and their

roles.

As for the conclusions of the study, although very well presented and structured, it would be

appropriate, as a suggestion, for a slightly greater discussion of what was observed, not only

as political participation in this specific context, but also to explore some possibilities for

understanding the democratic process as observed. This can inspire paths for new works

and approaches in different contexts.

-

In response to your suggestion for a more comprehensive discussion in the

Conclusions section:

Our survey does not allow us to determine whether today’s association members will be

tomorrow’s local elected representatives. However, we note that within the current

municipality studied, several members of the municipal council have beenn or still are,

involved in local associations. This observation aligns with the findings from the Nevers and

Bages survey (Nevers and Bages 2008), which compares 436 towns with fewer than 2500

inhabitants in the Midi-Pyr ´en ´ees region and found an undeniable correlation between

mayors’ past involvement in associations and their subsequent rise to the presidency of the

municipality. Similarly, Malet’s study, which compares more than 50 communes of 500 to 800

inhabitants in 40 different French departments, indicates that ”80 % of association leaders in

these communes have a relationship with their municipality” (Malet 2009). These findings

suggest that, beyond the professional sphere, political activity can be understood as a social

activity where associations serve both as training grounds for developing political techniques

and as potential springboards toward the professionalization of politics.

Reviewer #2: The theoretical contributions of the article are clear and significant. The study

extends existing theories of social networks and political participation by applying them in a

novel context using advanced methodological tools. The use of hypergraph analysis to

capture complex, higher-order interactions in social networks is particularly innovative.

I must commend the authors for revisiting well-grounded theories, particularly Granovetter's

strength of weak ties and Burt's structural holes concepts, a practice that more political

scientists should adopt. I also think that the attempt to step away from self-reported data and

shift to real-world data from a single municipality has the potential to capture higher-order

interactions that survey designs and even traditional network analysis research may miss.

The authors provide convincing evidence, particularly regarding the contribution of

heterogeneous activities’ participation on political participation for political participation

among association members.

Thank you for your positive feedback and for recognizing the theoretical contributions and

innovative aspects of our study.

That being said, I have a few misgivings and suggestions which I believe the authors could

address in a revised version of the paper.

1. What was the selection criteria for the 2 organizations out of each association group?

-

 We have clarified the selection criteria in the Data Collection section.

We identified 23 associations that frequently offer activities in the municipality. We excluded

sport and art associations mainly attended by minors and those with fewer than 4 active

members. Among the remaining 17, we arbitrarily selected 10 voluntary associations, two

from each of the following categories : art, educational, environmental protection, sport,

social.

2. Why external motivation must be limited to neighbors and not include other external

forces, such as those I see active on the news (a farmers road blockade, someone running

for mayor, etc.), which I am not familiar with, but push me to do something.

-

We appreciate the reviewer’s valuable observation, and we have addressed this by

discussing the limitation of our study that does not consider the role of external forces

in the Conclusions section.

While our study emphasizes the role of social ties, we acknowledge that external

forces—such as media coverage, public demonstrations, or political campaigns—can also

play a crucial role in shaping political behavior. It's important to note that this study is based

on correlation rather than regression analysis, which is more commonly used in surveys.

Although it would be valuable to comparatively assess the impact of social ties versus other

external forces, doing so would require a different approach to data collection, such as a

broader survey. Our study, which involved in-depth interviews with only 23 individuals, is

limited in its scope.

3. The study is limited to a single municipality in France, which with its higher than average

retirees and farming, hampers generalizatio

---

## [Editor Report · Decision Letter 1]

23 Sep 2024

Political Participation and Voluntary Associations : A Hypergraph Case Study

PONE-D-23-42942R1

Dear Dr. Azaiez,

We’re pleased to inform you that your manuscript has been judged scientifically suitable for publication and will be formally accepted for publication once it meets all outstanding technical requirements.

Kind regards,

Hocine Cherifi

Academic Editor

PLOS ONE
---

## [Editor Report · Acceptance letter]

26 Sep 2024

PONE-D-23-42942R1 

PLOS ONE

Dear Dr. Azaiez, 

I'm pleased to inform you that your manuscript has been deemed suitable for publication in PLOS ONE. Congratulations! Your manuscript is now being handed over to our production team.

Kind regards, 

on behalf of

Professor Hocine Cherifi 

Academic Editor

PLOS ONE